# Mendelian randomization analyses clarify the effects of height on cardiovascular diseases

Daniel Hui[1], Eric Sanford[2], Kimberly Lorenz [3,4,5,6], Scott M. Damrauer[4,6,7], Themistocles L. Assimes[8,9], Christopher S. Thom[10,11], Benjamin F. Voight[3,4,5,6]*

1 Graduate Program in Genomics and Computational Biology, University of Pennsylvania, Philadelphia, PA, United States of America, 2 Medical Scientist Training Program, Perelman School of Medicine, University of Pennsylvania, Philadelphia, PA, United States of America, 3 Department of Systems Pharmacology and Translational Therapeutics, University of Pennsylvania, Philadelphia, PA, United States of America, 4 Department of Genetics, Perelman School of Medicine, University of Pennsylvania, Philadelphia, PA, United States of America, 5 Institute for Translational Medicine, Perelman School of Medicine, University of Pennsylvania, Philadelphia, PA, United States of America, 6 Corporal Michael Crescenz VA Medical Center, Philadelphia, PA, United States of America, 7 Department of Surgery, Perelman School of Medicine, University of Pennsylvania, Philadelphia, PA, United States of America, 8 VA Palo Alto Health Care System, Palo Alto, CA, United States of America, 9 Department of Medicine, Stanford University School of Medicine, Stanford, CA, United States of America, 10 Division of Neonatology, Children's Hospital of Philadelphia, Philadelphia, PA, United States of America, 11 Department of Pediatrics, Perelman School of Medicine, University of Pennsylvania, Philadelphia, PA, United States of America

* bvoight@upenn.edu

**Data Availability Statement:** All data are publicly available, including: http://www.nealelab.is/uk-biobank, https://www.ebi.ac.uk/gwas/ Relevant

## Abstract

An inverse correlation between stature and risk of coronary artery disease (CAD) has been observed in several epidemiologic studies, and recent Mendelian randomization (MR) experiments have suggested causal association. However, the extent to which the effect estimated by MR can be explained by cardiovascular, anthropometric, lung function, and lifestyle-related risk factors is unclear, with a recent report suggesting that lung function traits could fully explain the height-CAD effect. To clarify this relationship, we utilized a well-powered set of genetic instruments for human stature, comprising >1,800 genetic variants for height and CAD. In univariable analysis, we confirmed that a one standard deviation decrease in height (~6.5 cm) was associated with a 12.0% increase in the risk of CAD, consistent with previous reports. In multivariable analysis accounting for effects from up to 12 established risk factors, we observed a >3-fold attenuation in the causal effect of height on CAD susceptibility (3.7%, p = 0.02). However, multivariable analyses demonstrated independent effects of height on other cardiovascular traits beyond CAD, consistent with epidemiologic associations and univariable MR experiments. In contrast with published reports, we observed minimal effects of lung function traits on CAD risk in our analyses, indicating that these traits are unlikely to explain the residual association between height and CAD risk. In sum, these results suggest the impact of height on CAD risk beyond previously established cardiovascular risk factors is minimal and not explained by lung function measures.

analysis code can be found at: https://github.com/daniel-hui/Height-CAD_MVMR.

**Funding:** This research is based on data from the Million Veteran Program, Office of Research and Development, Veterans Health Administration, and was supported by the Veterans Administration (VA) Cooperative Studies Program (CSP) award #G002. B.F.V. acknowledges support from the National Institutes of Health (DK126194 and DK101478) and Linda Pechenik Montague Investigator Award. C.S.T acknowledges support from the National Institutes of Health (HL156052). S.M.D. is supported by the U.S. Department of Veterans Affairs award IK2-CX001780. This publication does not represent the views of the Department of Veterans Affairs or the United States Government. The funders had no role in study design, data collection and analysis, decision to publish, or preparation of the manuscript.

**Competing interests:** D.H., E.S., K.L., T.L.A., C.S. T., and B.F.V. have no conflicting interest to report. S.M.D receives research support from RenalytixAI and personal consulting fees from Calico Labs, outside the scope of the current research. This does not alter our adherence to PLOS ONE policies on sharing data and materials.

# Introduction

Epidemiological evidence suggests that shorter height is associated with an increased risk of coronary artery disease (CAD), even after adjustment for known risk factors such as smoking status, lipid levels, body mass index (BMI), systolic blood pressure, and alcohol consumption [1]. Efforts to ascertain true effects of genetically influenced height on human health are important. Determinants of height include modifiable nutritional and socio-economic characteristics [2], as well as closely correlated cardiovascular or anthropometric traits (e.g., wider arteries or larger lungs). If specific assumptions hold (i.e., relevance, independence, and exclusion restriction), Mendelian randomization (MR) is a technique that can estimate causal relationships between traits by using genetic variants and their corresponding effects identified and measured through large-scale genetic analyses such as genome-wide association studies (GWAS). Studies using Mendelian randomization have provided evidence for a univariable causal association between shorter stature and risk of CAD [3, 4]. However, it is still unclear how much of this effect can be explained by established, conventional CAD risk factors. Previous work has suggested that the effect of height on CAD is nearly entirely explained by lung function, specifically forced expiratory volume in 1 second (FEV1) and forced vital capacity (FVC) [5]. This finding is backed by epidemiological evidence [6, 7], but results from other published univariable MR analyses are inconclusive [8].

Multivariable Mendelian randomization (MVMR) allows for the investigation of causal exposure-outcome trait relationships after accounting for additional factors that may confound or mediate simple, direct exposure-outcome associations. Here, we reaffirm the univariable causal relationship between height and CAD risk using recent GWAS for height and CAD with increased power. Using MVMR methods, we then investigated direct effects of height on susceptibility to CAD after accounting for the effects of 12 established risk factors. We also expanded these experiments to consider effects of height additional cardiovascular traits within this MVMR framework. Lastly, we investigated the effects of lung function on CAD risk using univariable and multivariable MR analyses, including repeating the exact procedures used in previous work with updated, publicly available datasets.

# Methods

## Variants used as instruments for height

We begin with the set of conditionally independent genome-wide significant ($p < 5 \times 10^{-8}$) SNPs [9] from a recent height GWAS [10] as the starting point for the construction of our genetic instruments for height. For robustness across exposures without conditionally independent results, only SNPs with genome-wide significance in both the marginal and conditional analysis were retained. To this initial pool of SNPs, we included an additional set of low frequency (1% to 5% MAF) or rare (0.1–1% MAF) genome-wide significant associations for height SNPs [11], where the SNP with lowest p-value was retained in case of duplicates. Proxies for palindromic SNPs (i.e., variants with A/T or C/G alleles) with $r^2 \geq 0.95$ were obtained from the primary height GWAS using 1000 Genomes EUR as LD reference [12, 13]–all variants were lifted over to hg19 (**URLs**). Finally, SNPs in networks of LD with $r^2 > 0.05$ were pruned using a graph-based approach with a greedy approximation algorithm (**URLs**) using 1000 Genomes Phase 3 EUR as the LD reference. This filtering resulted in an overall set of instruments which included 2,041 genome-wide significantly associated, non-palindromic, statistically independent ($r^2 < 0.05$) SNPs, including variants down to 0.1% allele frequency, nearly 2.5-fold more than previous efforts [5]. Of these 2,041 SNPs, 2,037 were present in the CAD GWAS and used in further analyses. Alleles and effect sizes were oriented to the height-

increasing allele. This set of instruments had 80% power to detect an odds ratio of 1.018 at alpha = 0.05 (**URLs**) [14], assuming a CAD GWAS sample size of 547,261, case ratio of 0.224, and $R^2$ of 0.264 in height using the height instruments. The authors did not have access to information that would identify any subjects for these data.

### Established cardiovascular risk factors

We obtained a collection of established risk factors for CAD from the literature to evaluate potentially indirect effects of height on CAD. Summary statistics from GWAS were acquired for body mass index (BMI) [10], plasma lipid levels (low density lipoprotein and triglycerides) [15], systolic and diastolic blood pressure [16], waist-hip ratio [17], smoking initiation [18], alcohol consumption (drinks per week [18]), educational attainment [19], birth weight [20], physical activity [21], and type 2 diabetes [22]. Units for systolic blood pressure were converted to 5 mmHg from 1 mmHg [16, 23]; the remaining GWAS were already expressed in units of standard deviations. For each trait, all alleles were harmonized to correspond to the height-increasing effect allele. The authors did not have access to information that would identify subjects from these data sets.

### Generation of additional instrument sets for FEV1 and FVC

GWAS summary statistics were obtained for FEV1 and FVC [24]. Lists of conditionally independent genome-wide significant lead variants were obtained from their respective GWAS, with palindromic SNPs and linkage disequilibrium addressed in an analogous way to the set of height instruments were filtered, as described above. 171 and 128 SNPs were included in the final set of instruments for FEV1 and FVC, respectively. Alleles and effect sizes were harmonized to the lung trait allele associated with higher trait values.

### Epidemiologic associations

Observed epidemiologic effects were manually curated from publicly available sources, including Wormser *et al*. [25] for height on ischemic stroke risk, Carter *et al*. [26] for height on abdominal aortic aneurysm risk, and Lai *et al*. [27] for all other outcomes.

### Mendelian randomization methods

Mendelian randomization studies rely on several assumptions to make valid conclusions. Valid independent genetic instruments must be associated (relevant) with the exposure trait without sharing a common cause with the outcome (independence) or affecting the outcome except through the risk factor (exclusion restriction) [28]. The experiments described herein met these assumptions, and were deemed to be valid instruments. Univariable two-sample Mendelian randomization analyses were carried out using the inverse-variance weighted (random effects model), weighted median [29], and MR-Egger [30] methods with the R package MendelianRandomization version 0.5.0 [31], with and without outlier removal using MR-PRESSO [32]. For completeness, we also present the results from eight additional methods (and note qualitatively similar results as to what we highlighted in the main text). In the univariable analysis, 2,037 of 2,041 SNPs for the height instrument were present in the CAD GWAS results and were used for analysis. Evidence of horizontal pleiotropy was evaluated using the intercept from MR-Egger. Outcome GWAS summary statistics were obtained from studies listed in **S1 Table** [15–21, 33–36].

Multivariable Mendelian randomization analysis was conducted using the R package WSpiller/MVMR [37]. Across all exposures, 1,906 SNPs of the 2,037 were present in every

GWAS and were used for analysis. Forest plots were constructed using the forestplot R package. Where indicated, odds ratios were converted to true effects by exponentiation [38].

## Results and discussion

We provide an overview of the experiments performed (**Fig 1**). We began by assembling GWAS data for height and CAD to perform experiments using the conventional, two-sample MR design. We utilized large, well-powered genetic studies for height and CAD, with mean sample size of 693,529 individuals per variant for height [10] and 122,733 CAD cases [39] (**S1 Table**). For height, additional low frequency (1–5% MAF) or rare (0.1–1% MAF) genome-wide significant SNPs with effects up to 2 cm/allele were also included [11]. After quality control and filtering (**Methods**), the final set of height instruments comprised of 2,037 variants,

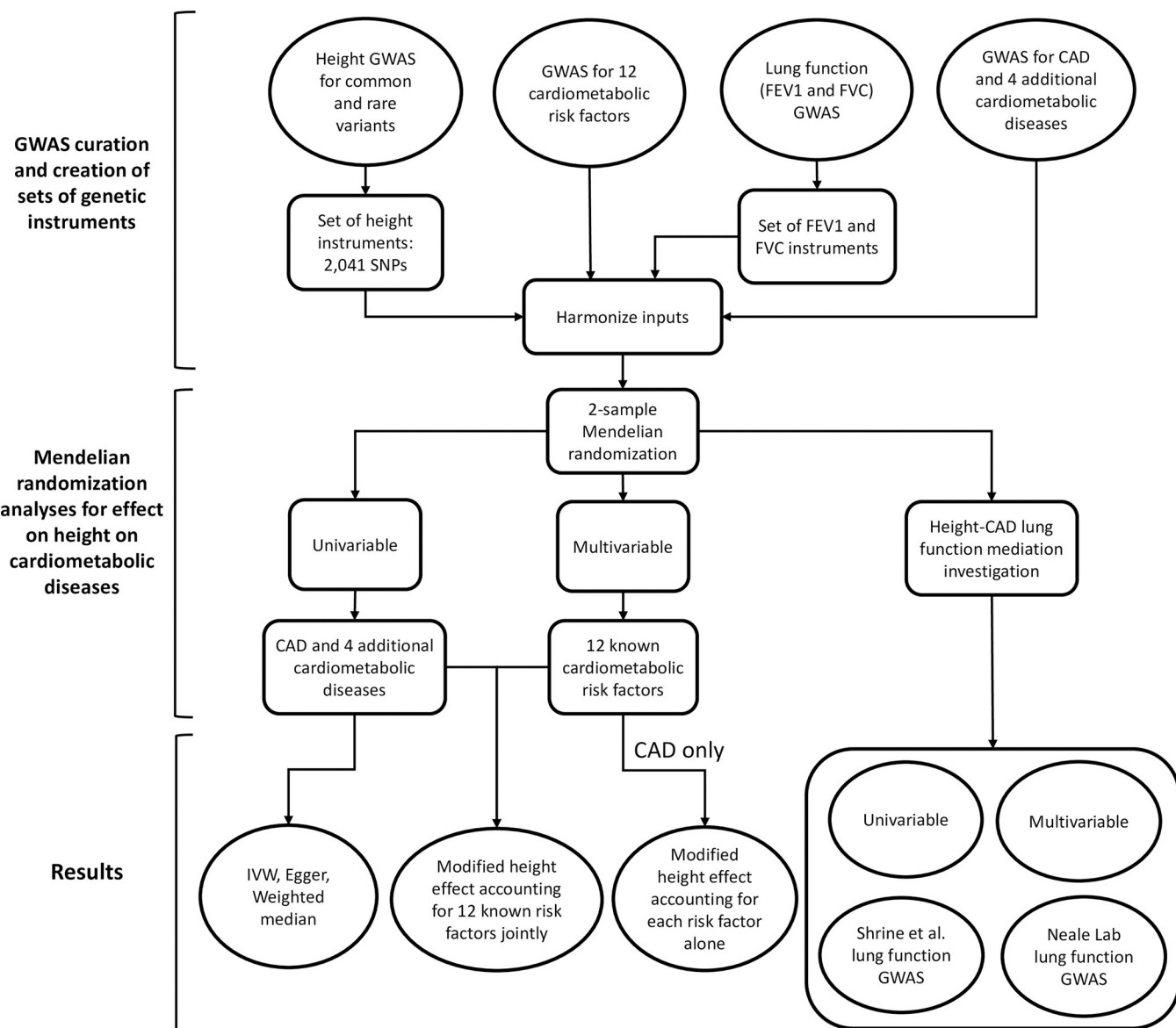

**Fig 1. Schematic flowchart outlining our experimental setup.** We collected genetic instrument for height and effect sizes for 12 additional CAD risk factors as well as for lung function. These data were used in two-sample Mendelian randomization in univariable and multivariable investigation.

nearly two and a half times that of previous efforts [5] (**S2 and S3** Tables), explaining a theoretical 26.4% of the variance in height (**S4 Table**). The set of variants had a Cragg-Donald F-Statistic of 121.5 [40] (based on the theoretical variance explained assuming Hardy-Weinberg equilibrium), minimizing any concerns regarding weak instrument bias [41, 42].

## Genetically influenced height is inversely associated with CAD risk

We next utilized these summary statistics to conduct univariable, two-sample MR of height on CAD risk (**Methods**). Using the inverse-variance weighted (IVW) method, we observed that a one standard deviation (SD) decrease in height (~6.5cm) [43] was associated with a 11.7% increase in the risk of CAD (OR = 1.12, 95% CI = 1.10–1.16, p = 1.6 x $10^{-20}$, **Fig 2**), a result that was consistent with previous reports [3–5]. This effect had persistent and robust association in sensitivity MR analyses, including weighted median (WM) (OR = 1.12, 95% CI = 1.09–1.15, p = 4.2 x $10^{-14}$) and MR-Egger (OR = 1.10, 95% CI = 1.04–1.16, p = 2.7 x $10^{-4}$) as well as other approaches (**S5 Table**). We further noted that the intercept estimated from MR-Egger was not significant, suggesting no direct statistical evidence supporting the presence of horizontal pleiotropy (p = 0.36). Analyses run using MR-PRESSO removed significant outliers, and returned similar effect estimates (**S6 Table**). Taken collectively, these results confirm the previously observed statistical effect of shorter stature with increased susceptibility to CAD. However, these results do not themselves address the extent to which height modulates CAD risk beyond the set of conventional established risk factors for CAD, which includes alcohol use [44], birth weight [45], body mass index [46], blood pressure [47], educational attainment [48], lipid levels [49], physical activity [50], smoking behavior [51], type 2 diabetes [52], or waist-hip ratio [53].

## Multivariable analysis substantially attenuates the association between genetically influenced height and CAD risk

We next sought to quantify the extent to which the catalog of established CAD risk factors could explain the association between lower stature and elevated risk of CAD. For this, we

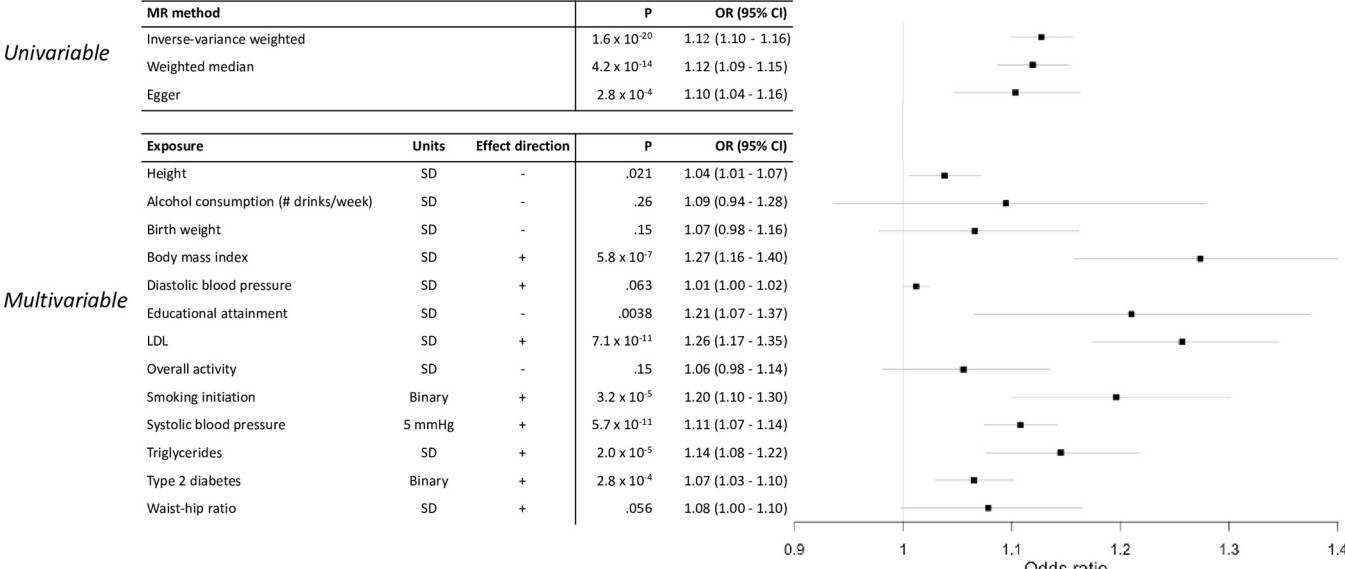

**Fig 2. Forest plots of two-sample MR results using height and 12 additional CAD risk factors.** a) Univariable height-CAD analyses using inverse-variance weighted, weighted median, and Egger MR methods. b) Multivariable MR results modeling all risk factors together. Effect direction indicates direction of the exposure that increases CAD risk–effects of all risk factors were flipped to OR>1.

utilized multivariable Mendelian randomization (MVMR) to account for effects of 12 risk factors with established or compelling associations with CAD (**Methods**). After obtaining recent, large-scale GWAS for all traits [10, 15–22], 1,906 of our 2,037 (93.6%) of variants were present in GWAS summary data across all traits and were utilized for MVMR analysis (**S7 Table**). We note that a reanalysis of this slightly smaller (N variants = 1,906) instrument returned virtually identical results to the above univariable results (IVW OR = 1.13, CI = 1.10–1.16, p = 4.7 x $10^{-19}$). Furthermore, the Sanderson-Windmeijer F statistic of the model was 40.9, indicating low bias due to weak instruments [37]. After including all 12 risk factors jointly with height, we observed that one SD shorter stature was associated with a 3.7% increase in the risk of CAD (OR = 1.04, 95% CI = 1.01–1.07, P = 0.021, **Fig 2, S8 Table**), representing a 68% attenuation in the effect compared to univariable MR. These results indicate a small, but significant, residual effect of height on susceptibility to CAD after accounting for known risk factors.

We next quantified the effects of each risk factor individually to explore the risk factors attenuating the effects of height on CAD risk in more detail. These effects were considered independent of the effects observed when all risk factors were jointly analyzed. In a series of MVMR models for height with each risk factor separately, we observed that 9 risk factors reduced height's effect on CAD risk, ranging between 0.06–3.65% of the total effect (**S9 Table**), somewhat consistent with previous work reporting 1–3% attenuation in height effect for each risk factor [5]. In contrast, 3 risk factors (smoking initiation, physical activity, and waist-hip ratio) increased the effect of height on CAD susceptibility, between 0.09%– 1.03%. These results indicate that the attenuation of the effect of height on CAD in the full joint model is not entirely explained by a single factor, but instead by a combination of risk factors in aggregate.

## Genetically determined lung function measures are not causally associated with susceptibility to CAD

Recent work suggested that lung function measures are inversely associated with cardiovascular events [6–8]. Furthermore, Mendelian randomization analyses suggested that the bulk of the causal effect of height on CAD susceptibility is complicated by FEV1 and FVC [5]. Given these reports, we next focused our efforts on (i) characterizing what relationship exists between these measures and CAD susceptibility, and (ii) if these measures could explain the residual putative causal effect estimated between height and CAD.

To develop sets of FEV1 and FVC genetic instruments and perform MVMR analyses, we utilized a large, recently published GWAS study for both traits [24]. After quality control and filtering analogous to what we applied to our height genetic instrument (**Methods**), the final sets of genetic variants comprised 171 for FEV1 and 128 for FVC, which explained a theoretical 3.26% and 2.32% of the genetic variance to each trait, respectively (**S10–S13 Tables**). These sets of instruments had Cragg-Donald F-Statistics of 78.8 and 74.1, respectively, arguing against weak instrument bias [40–42].

We then performed 5 sets of MR experiments to examine the effects of FEV1 and FVC, and their roles as attenuators of height's effect on susceptibility to CAD. Analyses were conducted using both univariable and multivariable methods, as well as recently published and legacy data sets. Collectively, we observed little attenuation of height's effect on CAD risk due to lung function factors, as well as scant univariable association between either lung function measure and CAD risk (**Fig 3, S14–S21 Tables**).

First, we performed univariable MR analysis for each lung function measure on CAD susceptibility (**Fig 3A**). We observed no significant effect of FEV1 on CAD risk using inverse-variance weighted, weighted median, or MR-Egger methods ($p_{IVW}$ = 0.08, $p_{WM}$ = 0.78, $p_{Egger}$ =

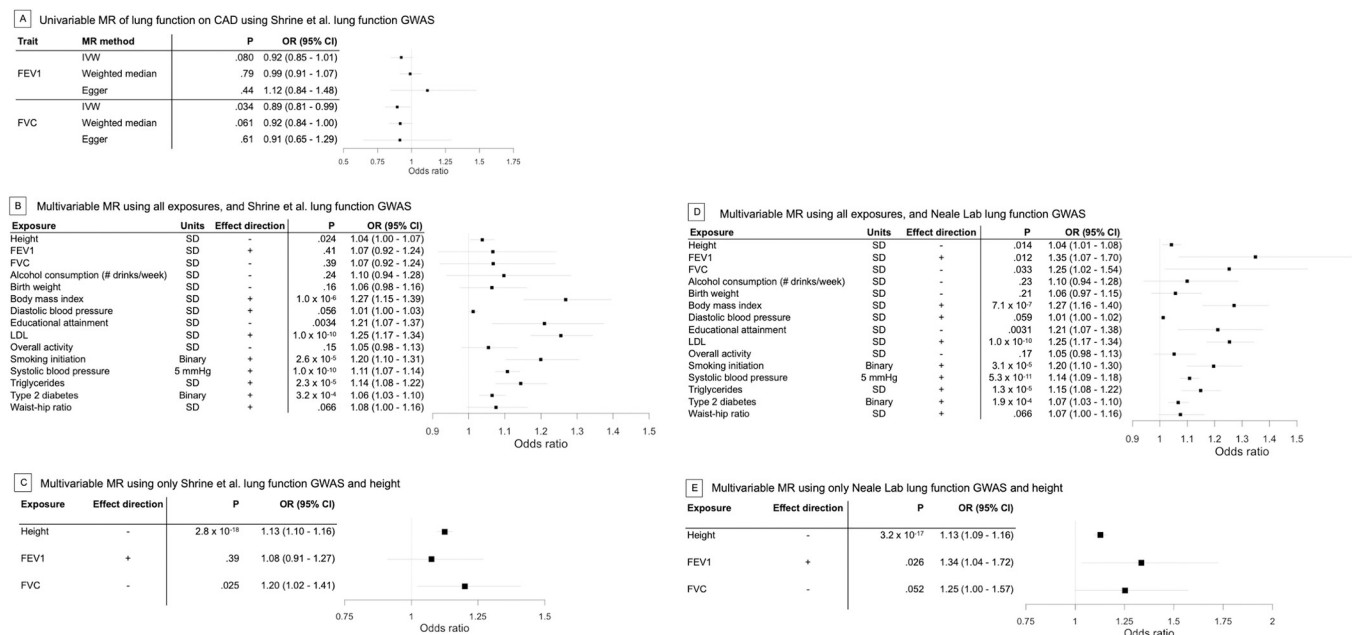

**Fig 3. Investigation of lung function in complicating the effect on CAD and a causal height-CAD relationship.** a) Univariable MR analyses of lung function and CAD. b) Multivariable MR results modeling all risk factors together, using Shrine et al. lung function GWAS. c) Multivariable MR results modeling height and lung function factors only using Shrine et al. lung function GWAS, on CAD risk. d) Multivariable MR results modeling all risk factors together, but using Neale Lab lung function GWAS. e) Multivariable MR results modeling height and lung function factors only but using Neale Lab lung function GWAS, on CAD risk.

0.44). Effects were directionally inconsistent, with increased FEV1 increasing CAD risk by MR-Egger, but increased FEV1 associated with decreased CAD risk by IVW and WM methods, though we note that all results do not reject the null hypothesis. For FVC, results were also not significant, except for those from the IVW method ($p_{IVW}$ = 0.034, $p_{WM}$ = 0.061, $p_{Egger}$ = 0.61). However, these results were directionally consistent across methods, with decreased FVC associated with increased CAD risk, consistent with epidemiological evidence [6, 7].

Second, we performed MVMR with FEV1 and FVC, both with and without the previously described 12 risk factors using the same 1,906 variants as were used in MVMR analysis. After including FEV1 and FVC, we observed a virtually identical, attenuated association between decreased height and CAD susceptibility as we observed from our prior experiments considering those 12 established risk factors without FEV1 and FVC (OR = 1.04, 95% CI = 1.00–1.07, p = 0.024, **Fig 3B**)**.** Neither FEV1 or FVC were associated with CAD susceptibility in this joint model ($p_{FEV1}$ = 0.41 and $p_{FVC}$ = 0.39). Similarly, when we excluded the 12 established risk factors and included only FEV1 and FVC as additional factors with height, we did not observe attenuation versus univariable height-CAD analysis (OR = 1.13, 95% CI = 1.10–1.16, p = 2.8 x $10^{-18}$, **Fig 3C**). Again, the associations for either lung function factor were, at best, modest ($p_{FEV1}$ = 0.39 and $p_{FVC}$ = 0.025).

Third, we performed additional MVMR analyses that instead utilized the lung function GWAS considered in prior analyses, which had suggested a lung function complicating effect (**URLs**). Again, we did not observe substantial attenuation in height's effect on CAD risk, (OR = 1.04, 95% CI = 1.01–1.08, p = 0.014, **Fig 3D**). When we modeled FEV1 and FVC exclusively with height, the effect of height on CAD risk did not appear to be reduced by lung function (OR = 1.13, 95% CI = 1.10–1.16, p = 3.2 x $10^{-17}$, **Fig 3E**), although the effects for both lung function measures were nominally significant at best ($p_{FEV1}$ = 0.026 and $p_{FVC}$ = 0.052)

Fourth, we created a restricted set of height genetic instruments that excluded variants nominally associated with FEV1 or FVC [24] (p < 0.05) to more directly compare between results from the published work suggesting lung function's effect on height [5], and only retained variants present in lung function, height, and CAD GWAS. Of 2,037 SNPs in our original set of height instruments, 1,112 remained and were used in univariable MR experiments analyzing the effect of height on CAD. We continued to observe a robust association between decreased height and increased CAD risk. Estimated effects, using WM (OR = 1.12, CI = 1.07–1.17, p = $1.0 \times 10^{-7}$) and IVW methods (OR = 1.13, CI = 1.09–1.17, p = $3.3 \times 10^{-11}$), were similar to our univariable analyses (**S1A Fig**), and the estimated effect from MR-Egger increased slightly (OR = 1.13, CI = 1.04–1.23, p = $2.7 \times 10^{-3}$).

Fifth, we filtered nominally associated FEV1 and FVC variants from our height IVs using older summary statistics for lung function (**URLs**) rather than Shrine et al [24], leaving 747 variants. In this case, we do observe an attenuation of association for height on CAD risk, consistent with the previous reports [4] (**S1B Fig**). Estimates of the effect of height on CAD risk also decreased (IVW estimate 12.0% to 8.9%, p = $1.6 \times 10^{-20}$ to p = $1.4 \times 10^{-3}$; WM estimate 11.3% to 3.7%, p = $4.2 \times 10^{-14}$ to p = 0.22; MR-Egger estimate 9.8% to 7.9%, p = $2.7 \times 10^{-4}$ to p = 0.24).

Finally, we performed mediation analysis to evaluate the direct relationship of height on CAD risk through lung function measures [54]. Compared to the UVMR effect (OR = 1.12), we found that the estimated expected direct effect was virtually identical after considering FEV1 and FVC (both ORs were = 1.12). This analysis was consistent with the above analyses, namely, the limited role of lung function measures on CAD risk.

### The effect of height on coronary artery disease risk and ischemic stroke risk, but not other cardiovascular disease risks, is attenuated by established risk factors

Finally, we extended our analyses to consider risk to other cardiovascular diseases. UVMR indicated that higher stature was associated with reduced risk for ischemic stroke (similar to that observed from CAD). However, this result attenuated to the null after accounting for CVD risk factors via MVMR (OR: 1.03, CI: 0.99–1.08) **Fig 4**). The effects of height on other cardiovascular disease risks (venous thromboembolism, abdominal aortic aneurysm, atrial fibrillation) were not affected by accounting for CVD risk factors, remaining roughly consistent with epidemiologic observations and univariable MR estimations (**Fig 4**). Finally, consistent with our prior results, the effect of height on coronary artery disease risk was attenuated in a MVMR context adjusting for up to 12 established cardiovascular risk factors, negating the significant effects seen in epidemiologic and univariable studies (**Fig 4**).

### Conclusions

Using well-powered GWAS, we recapitulated previous evidence that lower stature increases susceptibility to CAD, estimating a 12% increased CAD risk per 1 standard deviation unit decrease in height contingent on standard MR assumptions (**Methods**). This effect was substantially attenuated to 3.7% after accounting for a collection of 12 CAD risk factors. In contrast with prior work, we demonstrate–through univariate and multivariable MR experiments directly–that lung function explains little, if any effect, of height on CAD susceptibility. Our findings refute the notion that height provides an independent and substantial effect on CAD risk beyond contributions from other risk factors [2].

Several factors could be driving differences between our work and previously published findings of lung function on CAD risk. Certain phenotype-specific covariates, namely height,

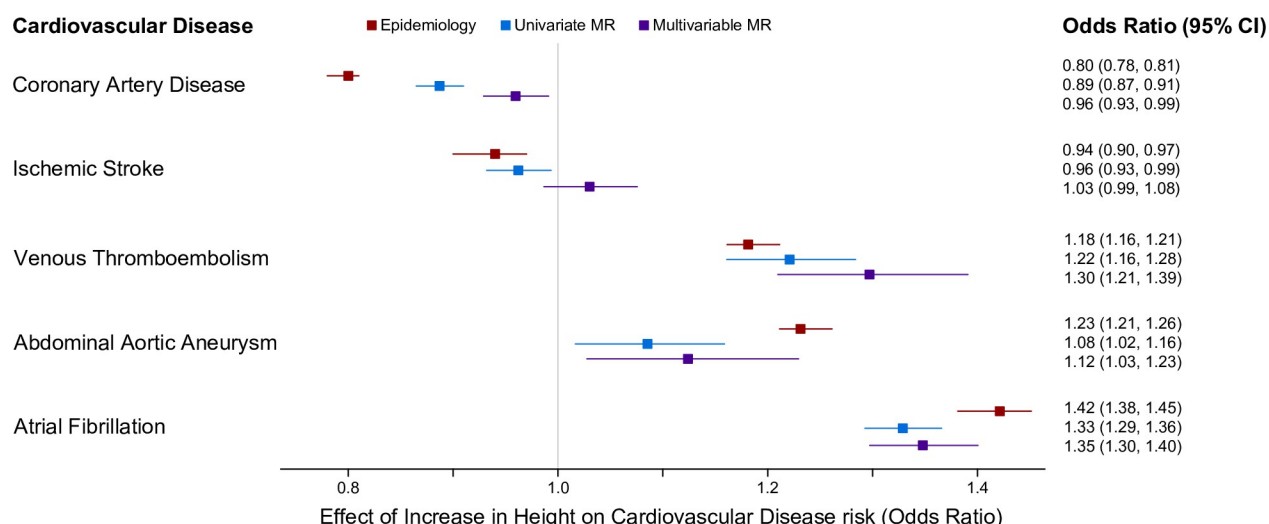

**Fig 4. Effects of height on cardiovascular disease risks.** Multivariable MR including 12 additional cardiovascular disease risk factors as exposures attenuates the effect of height on coronary artery disease risk and ischemic stroke, but does not impact the effects of height on venous thromboembolism, abdominal aortic aneurysm, or atrial fibrillation.

were not included in the lung function GWAS results used in Marouli et al. [5]. By contrast, the GWAS we used included height as a covariate in the models used for association testing. It is known that including heritable covariate-adjusted data can induce collider biases in MR studies [55]. While this might cloud the interpretation of the results using those data for univariate and multivariable experiments about the relationship between FEV1 and/or FVC and CAD risk, our major conclusion contrasts previous reports. In our experiments, lung function did not explain the relationship of height to CAD. In multivariable analyses, a small effect of height persisted (**Fig 3D**). One reasonable interpretation of these findings is that lung function may be a causal risk factor for CAD as reported previously [56], even when considered jointly with other cardiometabolic risk factors.

We note that the procedure used in Marouli et al., in which variants nominally associated (i.e., p < 0.05) with lung function were removed from height instruments, inherently reduces power. This reduction could have influenced the apparent attenuation of lung function on height's effect on CAD risk. However, we did not observe a similar attenuation when we utilized their procedure using a larger genetic instrument for height. These findings occurred despite strong similarities between lung function GWAS size (Shrine et al. N = 400,102 vs. Neale Lab N = 361,194) and high degree of sample overlap, with both GWAS being primarily conducted in the UK Biobank. The collection of these results provide evidence against the hypothesis that lung function substantially attenuates height's genetically causal effect on CAD risk. Instead, other established CAD risk factors attenuated the relationship between height and CAD.

One exposure we considered in the multivariable analysis but did not include was high-density lipoprotein (HDL) cholesterol levels. While epidemiological evidence has suggested an inverse relationship between HDL and CAD risk [57], randomized controlled trials and MR studies have largely not supported this relationship [58, 59], and newer evidence has suggested the role of HDL on CAD risk is nonlinear and/or context specific [60]. Moreover, HDL is correlated with triglyceride levels, which does have a growing body of evidence to support causality and which we did include in our multivariable analysis. Thus, we opted to not include HDL in these analyses. Furthermore, while not significantly different from OR = 1, the relationship

between alcohol exposure and CAD is complex and potentially non-linear [44], and thus we advise strong caution against strong interpretations of the directionality of alcohol exposure in multivariable analyses. Future work could consider ways to examine heterogeneity within genetically instrumented height, to potentially future explore how mediating factors (in the context of height) could matter for CVD endpoints.

Our study had some limitations. First, there is some sample overlap between exposure and outcome traits, which may induce bias in effect estimates [61]. However, given the samples sizes underlying these data sets, we expect the effect of biases from sample overlap to be minimal. Second, we were limited to the extent that we could address residual population stratification and/or assortative mating not accounted for in the primary GWAS. Future MR studies using siblings could potentially address this limitation [62], with prior work demonstrating stronger effects of height on CAD risk using this design [63]. Finally, how well these findings may translate to non-European ancestries remains unclear as our analyses were conducted almost exclusively in individuals of predominantly European descent [64–66]. Even within ancestry groups, stratification effects–such as population structure and stratification across Europe [67]–or context-specificity [68], evident in BMI [69, 70], may cause genetic effects to vary. Future work to explore differences in risk by ancestry would be an important consideration as such data become more readily available. We included BMI in our multivariable experiments, which includes of course Height as a correction for weight; with individual level data, this might induce multi-collinearity issue which are avoided here through the use of summary statistics. This makes the interpretation of the height coefficient dependent on the BMI coefficient in this case. We note that if we exclude BMI from MVMR entirely, attenuation of the effect on Height to CAD does persist (= -0.065), though this does not entirely account for the effects of obesity at height loci, and we know obesity is a risk factor for CVD.

In conclusion, we observed a limited effect of height on CAD after accounting for established risk factors. We additionally provide evidence that lung function has a minimal role in attenuating the effect of height on CAD, which contrasts recently published work. These findings help explain the clinical and epidemiological significance of height as a risk factor for CAD and other cardiovascular conditions.

## URLs

UCSC Liftover: https://genome.ucsc.edu/cgi-bin/hgLiftOver

Select analysis code: https://github.com/daniel-hui/Height-CAD_MVMR

Neale Lab UK Biobank GWAS results: http://www.nealelab.is/uk-biobank

MR power calculation: https://shiny.cnsgenomics.com/mRnd

## Supporting information

**S1 Checklist. STROBE-MR checklist.**
(DOCX)

**S1 Fig.** Univariable effect of 1-SD decrease in height on CAD risk after removing variants from set of height instruments that are nominally associated ($p < 0.05$) with lung function from GWAS conducted by: (a) Shrine et al. (N SNPs = 1,112) and (b) Neale Lab (N SNPs = 747).
(TIF)

**S1 Table. Information on all genome-wide association summary statistics used for this study.**
(XLSX)

**S2 Table. All variants and their effects used for the genetic instrument for height.**
(XLSX)

**S3 Table. LD proxies used for strand ambiguous SNPs in set of height instruments.**
(XLSX)

**S4 Table. Cragg-Donald F statistics for instrument sets.**
(XLSX)

**S5 Table. Univariable Mendelian randomization results for height-CAD.**
(XLSX)

**S6 Table. Univariable Mendelian randomization results for height and lung function exposures on CAD after MR-PRESSO outlier removal.**
(XLSX)

**S7 Table. Variants and their effects used in Multivariable Mendelian randomization analyses.**
(XLSX)

**S8 Table. Multivariable Mendelian randomization results using height and 12 additional risk factors in one joint model.**
(XLSX)

**S9 Table. Multivariable Mendelian randomization results using height and each of the 12 additional risk factors in separate models.**
(XLSX)

**S10 Table. All variants and their effects used in set of FEV1 instruments.**
(XLSX)

**S11 Table. LD proxies used for strand ambiguous SNPs in set of FEV1 instruments.**
(XLSX)

**S12 Table. All variants and their effects used in set of FVC instruments.**
(XLSX)

**S13 Table. LD proxies used for strand ambiguous SNPs in set of FVC instruments.**
(XLSX)

**S14 Table. Univariable FEV1-CAD MR results using data from Shrine et al.**
(XLSX)

**S15 Table. Univariable FVC-CAD MR results using data from Shrine et al.**
(XLSX)

**S16 Table. Multivariable Mendelian randomization results using height, 12 additional risk factors, and FEV1 and FVC effects from Shrine et al. in one joint model.**
(XLSX)

**S17 Table. Multivariable Mendelian randomization results height and lung function effects from Shrine et al.**
(XLSX)

**S18 Table. Multivariable Mendelian randomization results using height, 12 additional risk factors, and FEV1 and FVC effects from UKBB GWAS (Neale Lab), in one joint model.**
(XLSX)

**S19 Table. Multivariable Mendelian randomization results height and lung function effects UKBB GWAS (Neale Lab).**
(XLSX)

**S20 Table. Univariable height-CAD MR results after removing p<0.05 SNPs from Shrine et al. lung function GWAS.**
(XLSX)

**S21 Table. Univariable height-CAD MR results after removing p<0.05 SNPs from UKBB GWAS (Neale Lab) lung function GWAS.**
(XLSX)

## Acknowledgments

For data from Klarin et al, approved dbGAP access to phs001672.v6.p1 was provided to B.F.V. (dbGAP project ID: 27398). The authors thank the Million Veteran Program (MVP) staff, researchers, and volunteers, who have contributed to MVP, and especially participants who previously served their country in the military and now generously agreed to enroll in the study (see https://www.research.va.gov/mvp/ for more details). This publication does not represent the views of the Department of Veterans Affairs or the United States Government.

## Author Contributions

**Conceptualization:** Christopher S. Thom, Benjamin F. Voight.

**Data curation:** Daniel Hui, Benjamin F. Voight.

**Formal analysis:** Daniel Hui, Eric Sanford, Kimberly Lorenz, Benjamin F. Voight.

**Funding acquisition:** Benjamin F. Voight.

**Investigation:** Christopher S. Thom.

**Project administration:** Benjamin F. Voight.

**Resources:** Benjamin F. Voight.

**Supervision:** Benjamin F. Voight.

**Visualization:** Daniel Hui, Eric Sanford, Benjamin F. Voight.

**Writing – original draft:** Daniel Hui, Christopher S. Thom, Benjamin F. Voight.

**Writing – review & editing:** Daniel Hui, Kimberly Lorenz, Scott M. Damrauer, Themistocles L. Assimes, Christopher S. Thom, Benjamin F. Voight.

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
