## [Decision Letter · Decision Letter 0]

19 Jul 2023

PONE-D-23-13724Mendelian randomization analyses clarify the effects of height on cardiovascular diseasesPLOS ONE

Dear Dr. Voight,

Thank you for submitting your manuscript to PLOS ONE. After careful consideration, we feel that it has merit but does not fully meet PLOS ONE’s publication criteria as it currently stands. Therefore, we invite you to submit a revised version of the manuscript that addresses the concerns raised by reviewers, listed below.

We look forward to receiving your revised manuscript.

Kind regards,

Marie-Pierre Dubé, PhD

Academic Editor

PLOS ONE

Journal Requirements:

This research is based on data from the Million Veteran Program, Office of Research and Development, Veterans Health Administration, and was supported by the Veterans Administration (VA) Cooperative Studies Program (CSP) award #G002. B.F.V. acknowledges support from the National Institutes of Health (DK126194 and DK101478) and Linda Pechenik Montague Investigator Award. C.S.T acknowledges support from the National Institutes of Health (HL156052). S.M.D. is supported by the U.S. Department of Veterans Affairs award IK2-CX001780. This publication does not represent the views of the Department of Veterans Affairs or the United States Government. The funders had no role in study design, data collection and analysis, decision to publish, or preparation of the manuscript.

NO - The funders had no role in study design, data collection and analysis, decision to publish, or preparation of the manuscript.

D.H., E.S., K.L., T.L.A., C.S.T., and B.F.V. have no conflicting interest to report. S.M.D receives research support from RenalytixAI and personal consulting fees from Calico Labs, outside the scope of the current research.

Reviewers' comments:

Reviewer's Responses to Questions

**Comments to the Author**

1. Is the manuscript technically sound, and do the data support the conclusions?

Reviewer #1: Yes

Reviewer #2: Yes

2. Has the statistical analysis been performed appropriately and rigorously? 

Reviewer #1: Yes

Reviewer #2: Yes

3. Have the authors made all data underlying the findings in their manuscript fully available?

Reviewer #1: Yes

Reviewer #2: Yes

4. Is the manuscript presented in an intelligible fashion and written in standard English?

Reviewer #1: Yes

Reviewer #2: Yes

5. Review Comments to the Author

Reviewer #1: This is an interesting study on the effects of height on CAD and cardiovascular outcomes. The results confirm the causal effect of height on CAD, but this effect is significantly attenuated after conditioning on 12 epidemiological risk factors for CAD. The mediating role of lung function (FEV1 and FVC) is also explored in MVMR and univariate MR of lung function on CAD, and the null results contradict previous evidence. The results shed light on the association between height and CAD, and complement previous studies on the topic, both MR and epidemiological. Overall, the paper is well written and the methods are robust. My main concern is the overlap between GWAS populations tested in the two-sample MR setting of this study, which can lead to weak instrument or winner’s curse. This can be addressed using new MR methods that adjust IVW estimates taking into consideration sample overlap.

Major comments:

As mentioned above, I am concerned about overlap of samples in the exposure and outcome GWAS. In the discussion , the authors mention the overlap of samples among GWAS used in the two-sample MR, but could they should further comment on the impact of this on the MR findings. New methods are available and can be applied in case of sample overlap between GWAS used in a two sample MR context to attenuate bias ( see Mounier & Kutalik, 2023; PMID: 37036286).

In the same direction, in table S1, it is important to provide additional descriptives of the cohorts, for instance, mentioning if UKBB or other biobanks were part of the GWAS meta-analyses, and the N of samples of these biobanks participating in each GWAS meta-analysis- this could help the reader appreciate better the extend of the overlap. Also, given the importance of adjustment for variables such as height in the various GWAS of epidemiological risk factors, please add a column in Table S1 indicating for which covariates the GWAS phenotype was adjusted. Please discuss on the implications of the different covariates used in the studies ( see Hartwig, 2021; PMID: 33619569).

Results: is the reported F-statistic a mean F-statistic or a cumulative one? I would assume a much larger cumulative F-statistic.

The authors claim that there is a 12% increase in risk of CAD in the univariate MR analysis, but the actual IVW OR is 1.13, shouldn't that be 13% then?

It would be helpful to provide estimates of pleiotropy for the MR instruments- such as the MR-Egger intercept p-value, Cohran Q heterogeneity estimates, for all the univariate MR analyses- and comment on these. Also, given the increased number of SNP-instruments notably for height, additional sensitivity analyses such as the MR-PRESSO and the leave one out analysis could identify outlier SNPs and remove them. The same applies to MR studies using as exposure FEV1 or FVC.

Methods: The clump R2 of 0.05 used to define independent SNPs for height and FVC-FEV1 could be liberal- the default clumping R2 in the TwoSampleMR package is 0.001. The authors should consider a more stringent clumping R2.

Minor comments:

Introduction, last paragraph: there is the “factors” word missing at the end of a sentence: “12 established risk”. Also, in the same line, an “on” is missing after the “effects of height”.

The term legacy datasets is used but it infers that some datasets might be obsolete. I suggest replacing it with “publicly available” datasets.

Methods, paragraph : “Generation of additional instrument sets for FEV1 and FVC”: “Alleles and effect sizes were harmonized to the height-increasing effect allele”. Do the authors mean FEV1 or FVC-increasing alleles?

Paragraph “multivariate analysis”: last sentence, there is a typo: attenuation of the effect of height of CAD – it should be on CAD instead.

Lung function measures are not genetically causally associated with CAD susceptibility: this does not read well. Genetically determined lung function is not causally associated with CAD

Discussion: The authors state that ”Our findings refute the notion that height, height-related nutritional status, or socioeconomic status meaningfully contributes to CAD risk (4)”. Based on the findings of the present study, one can argue that height does not have an important contribution to CAD risk, but nutritional status and socioeconomic status could actually play a role (see Figure 2) based on the results of the MVMR- but independently from height.

In Figure 2. the direction of effect of educational attainment is towards an increasing risk of CAD. Have the authors aligned SNPs for educational attainment to a decreasing effect for this exposure? This should be made clear, since miss-interpretation can occur.

Please consider filling out the MR-STROBE ckecklist.

Reviewer #2: Hui et al. used a MR approach to evaluate the effect of height on coronary artery disease and other cardiovascular diseases. The authors use both univariable models and multivariable models adjusting for known CAD risk factors. They also consider a possible mediation of the effect of height on CAD by lung function phenotypes and reach the conclusion that there are no substantial mediating effect. This contrasts with previous reports that suggested that lung function was the major mediator of the height--CAD relationship. Overall, the report is clear and well written and the methodology is robust although I have some concerns listed below.

Major comments

- The set of variants used as an instrument for height explains a large proportion of the heritability, which is desirable to attenuate the risk of weak instrument bias. However, the bias due to violations of the exclusion restriction assumption (i.e. IV independant from outcome given exposure) are not discussed. In this study, ~2000 variants spanning a large portion of the genome are used and certainly have direct effects on CAD either through unmeasured pathways or through LD with CAD risk loci biasing the MR estimates. As a sensitivity analysis, instruments (robustly) associated with height and with few other anticipated phenotypic effects should be used. Ideally, the variants included in such an analysis could be argued, biologically, to have direct effects on height.

- In the multivariable models, body mass index is included. In my view, since BMI is calculated using height (and weight), this complexifies the interpretation of the results. For instance, an attenuation would be expected as part of the heigh effect may be attributed to the BMI coefficient. In an MR setting based on individual level data, this would lead to multicolinearity issues which is numerically avoided in the summary statistics setting. However, here, the height coefficient can't be interpreted independently from the BMI coefficient.

- Did the authors consider formal mediation analysis (e.g. Burgess IJE 2015) which seem appropriate to answer the question of mediation of height -> lung function -> cardiovascular disease.

Minor comments

- Given the large number of variants included as a genetic instrument of heigh and the fact that it's impossible to account for all possible biasing paths using pre-selected risk factors, the authors could have used unsupervised MR models that flexibly accounts for genetic effect heterogeneity. For example, the MR-Clust algorithm (Foley et al. Bioinformatics 2020) uses a mixture model to estimate MR effects that can vary. A post-hoc analysis of the inferred clusters of genetic instruments using summary statistics for known risk factors could then inform the main modulatory risk factors. I know that this approach is quite different from the current report, but perhaps it could be discussed in the future work or limitations section.

- Add number of participants from UKB in the summary statistics to estimate sample overlap (Sup Table 1). The analysis is presented as a two-sample MR, but as the authors note it is quite likely that the reliance on the UK Biobank leads to substantial sample overlap. Providing this information could help readers assess the degree of overlap.

- In the 1st paragraph of the discussion, the sentence "Our findings refute the notion that height, height-related nutritional status, or socioeconomic status meaningfully contributes to CAD risk" should be rephrased. In its current form, it reads as if socioeconomic status does not contribute to CAD risk. Moreover, the point estimate for the fully adjusted OR is 3.7% increase in the risk of CAD for a ~6.5cm decrease in height which could still be considered meaningful (especially for people that fall further away from the mean). I think this statement could be made more precise.

6. PLOS authors have the option to publish the peer review history of their article (what does this mean?). If published, this will include your full peer review and any attached files.

Reviewer #1: **Yes: **Despoina Manousaki

Reviewer #2: No

---

## [Author Response · Author response to Decision Letter 0]

11 Oct 2023

A detailed response to reviewers is included in the revision as an attached file.

---

## [Decision Letter · Decision Letter 1]

26 Oct 2023

PONE-D-23-13724R1Mendelian randomization analyses clarify the effects of height on cardiovascular diseasesPLOS ONE

Dear Dr. Voight,

Thank you for submitting your manuscript to PLOS ONE. After careful consideration, we feel that it has merit but does not fully meet PLOS ONE’s publication criteria as it currently stands. Therefore, we invite you to submit a revised version of the manuscript that addresses the points raised during the review process.

We thank you for your detailed response to the reviewer's comments. Reviewer #1 has raised additional observations that merit consideration, see comments below. Please respond to these additional comments. Rearding the comment on network MR analysis, we strongly encourage you to conduct the analysis as a additional sensitivity assessment. If you opt out of conducting the additional analysis, we recommend that you provide a discussion of this alternative analysis approach in the discussion, including its strengths, limitations and possible impact the results of this analysis would have had on the interpretation of your findings.

We look forward to receiving your revised manuscript.

Kind regards,

Marie-Pierre Dubé, PhD

Academic Editor

PLOS ONE

Journal Requirements:

Reviewers' comments:

Reviewer's Responses to Questions

**Comments to the Author**

1. If the authors have adequately addressed your comments raised in a previous round of review and you feel that this manuscript is now acceptable for publication, you may indicate that here to bypass the “Comments to the Author” section, enter your conflict of interest statement in the “Confidential to Editor” section, and submit your "Accept" recommendation.

Reviewer #1: All comments have been addressed

Reviewer #2: All comments have been addressed

2. Is the manuscript technically sound, and do the data support the conclusions?

Reviewer #1: Yes

Reviewer #2: Yes

3. Has the statistical analysis been performed appropriately and rigorously? 

Reviewer #1: Yes

Reviewer #2: Yes

4. Have the authors made all data underlying the findings in their manuscript fully available?

Reviewer #1: Yes

Reviewer #2: Yes

5. Is the manuscript presented in an intelligible fashion and written in standard English?

Reviewer #1: Yes

Reviewer #2: Yes

6. Review Comments to the Author

Reviewer #1: I thank the authors for responding to my previous comments.

I have some minor suggestions, which are outlined below.

In their current format, the Figures are almost illegible. Please provide a better resolution.

Abstract: "established risk factors" is a vague term. Could the authors be more precise and say instead cardiovascular, anthropometric, lung function, and lifestyle-related risk factors.

Line 77: “may complicate”: MVMR identifies confounding or mediating effects. I would suggest saying instead “confound or mediate”.

Could the authors explain why proxies were not sought, especially in the multivariable MR were there were more missing SNPs?

Line 128: “to the lung trait effect allele”. Do the authors mean that the alleles were harmonized to have an increasing effect on the lung trait?

Line 318: The authors here talk about other CV outcomes that the CAD, but start the paragraph with the statement on the CAD (which has been shown before). I suggest mentioning the results on ischemic stroke first (that’s the novelty of the analysis presented here), and then extend to the other tested CV outcomes, for which the results were non-significant.

Line 340: provides an independent and substantially: please correct grammar here (the word effect is missing).

Point 22 in the rebuttal. I agree with Reviewer 2 that a two-step network MR as described by Burgess et al would be an asset in the manuscript, since the aim here is really to explore mediation. I strongly recommend adding this analysis.

Reviewer #2: (No Response)

7. PLOS authors have the option to publish the peer review history of their article (what does this mean?). If published, this will include your full peer review and any attached files.

Reviewer #1: No

Reviewer #2: No

---

## [Author Response · Author response to Decision Letter 1]

4 Dec 2023

Responses to reviewers are provided in the rebuttal document.

---

## [Editor Report · Decision Letter 2]

31 Jan 2024

Mendelian randomization analyses clarify the effects of height on cardiovascular diseases

PONE-D-23-13724R2

Dear Dr. Voight,

We’re pleased to inform you that your manuscript has been judged scientifically suitable for publication and will be formally accepted for publication once it meets all outstanding technical requirements.

Kind regards,

Marie-Pierre Dubé, PhD

Academic Editor

PLOS ONE
---

## [Editor Report · Acceptance letter]

13 Feb 2024

PONE-D-23-13724R2 

PLOS ONE

Dear Dr. Voight, 

I'm pleased to inform you that your manuscript has been deemed suitable for publication in PLOS ONE. Congratulations! Your manuscript is now being handed over to our production team.

Kind regards, 

on behalf of

Dr. Marie-Pierre Dubé 

Academic Editor

PLOS ONE